

# (Almost) everything is a Dicke model – Mapping non-superradiant correlated light-matter systems to the exactly solvable Dicke model

**Andreas Schellenberger**[⋆] **and Kai Phillip Schmidt**[†]

Friedrich-Alexander-Universität Erlangen-Nürnberg (FAU), Department of Physics,
Staudtstraße 7, 91058 Erlangen, Germany

⋆ andreas.schellenberger@fau.de , † kai.phillip.schmidt@fau.de

## Abstract

We investigate classes of interacting quantum spin systems in a single-mode cavity with a Dicke coupling, as a paradigmatic example of strongly correlated light-matter systems. Coming from the limit of weak light-matter couplings and large number of matter entities, we map the relevant low-energy sector of a broad class of models in the non-superradiant phases onto the exactly solvable Dicke model. We apply the outcomes to the Dicke-Ising model as a paradigmatic example [1, 2], in agreement with results obtained by mean-field theory [2]. We further accompany and verify our findings with finite-size calculations, using exact diagonalization and the series expansion method pcst++ [3].

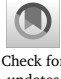

# 1   Introduction

Over the last decades, advancements in the field of cavity quantum electrodynamics as well as circuit quantum electrodynamics paved the way to study systems of matter strongly and collectively coupled to a light mode. These experimental breakthroughs made it possible to realize and study paradigmatic theoretical models like the Rabi, Tavis-Cummings, and Dicke model with strong light-matter interactions in the lab [4–11]. With these tools at hand, a fundamental question is how the interactions between light and matter do influence each other, altering properties of the separated (potentially complicated) individual parts, like observables, local interactions, or the location of phase transitions [12–22].

One of the paradigmatic light-matter systems is the Dicke model, with a minimal setup both on the light and the matter part [23, 24]. The model consists out of $N$ individual spin-1/2 particles that are individually coupled to a single cavity mode. Hepp and Lieb showed for the thermodynamic limit $N \to \infty$ that this system can be solved analytically by a Bogoliubov transformation and features a second-order phase transition from a normal to a superradiant phase with a non-vanishing photon density in the ground state [24]. While the matter part of the Dicke model consists out of an arbitrary number of spins, it breaks down to a non-interacting problem in the case of no light-matter interaction, as the local degrees of freedom are only coupled through the cavity, making it trivially solvable.

To make the composite system more interesting, various generalizations for the Dicke model were proposed and discussed, like more complex local spin structures [25], multi-mode cavities [24, 26, 27], non-Hermitian generalizations [28], open systems [29, 30], altered light-matter interactions [31, 32], non-equilibrium systems [33], and added matter-matter interactions between the spins [2, 34, 35]. One prime example is the Dicke-Ising model, where an additional Ising interaction between nearest-neighbor spins is present. Using mean-field theory and a classical approximation of the spin degrees of freedom, Zhang et al. found a rich phase diagram for antiferromagnetic interactions including superradiant phases, with both an antiferromagnetic and a paramagnetic phase in the matter part [2]. However, using quantitative numerical techniques, deviations were found for the phase transitions both in the location as well as in the order in 1D [1, 36].

In this work we elaborate on this line of thinking by considering a more generalized setting for the matter part, consisting of long-range hopping and correlated processes, and coupling it to a single light mode. This enables us to investigate the interplay between the correlations and effects induced by the light-matter and the matter-matter coupling. Restricting ourselves to the non-superradiant phases that are adiabatically connected to the case of vanishing light-matter interaction, we establish an analytical solution of the low-energy part of this model by mapping it to an effective Dicke model. This enables us to study the low-lying excitations of this generalized Dicke model in the non-superradiant phases analytically, including the closing of the gap, potentially inducing second-order phase transitions.

The structure of this paper is as following. First, in Sec. 2 we introduce the general framework, including the generalized model and the prerequisites to derive the effective Dicke model. The latter is done in the Subsecs. 2.2 and 2.3, first giving some physical intuition how the system can be solved, followed by a general derivation on operator level. In Sec. 3 we apply our general findings onto the Dicke-Ising model as an exemplary case. We compare our outcomes, obtained in the thermodynamic limit, with results from exact diagonalization (ED) and the series expansion method pcst$^{++}$ [3] on finite systems to strengthen the validity of the effective model. In Sec. 4 we draw conclusions and give an outlook for potential research directions.



# 2 Derivation of the effective Dicke model

In this section we will derive the main finding of this work on a general level. First, we will define the framework including the light-matter model and further conditions we assume. Thereafter, we present our two approaches to show the decoupling of the Dicke part and the rest of the Hamiltonian. While the second one is more rigorous and general, the first one helps understanding the decoupling in a more intuitive way being less technical.

## 2.1 General framework

For this work we concentrate on a general Hamiltonian $\mathcal{H}$ of the form

$$\mathcal{H} = \mathcal{H}_{\mathrm{D}} + \mathcal{H}_{\mathrm{matter}}, \tag{1}$$

where $\mathcal{H}_{\mathrm{D}}$ denotes the Dicke Hamiltonian [24] and $\mathcal{H}_{\mathrm{matter}}$ represents an Hamiltonian with additional matter-matter interactions. For the Dicke Hamiltonian we use the following notation

$$\mathcal{H}_{\mathrm{D}} = \frac{\omega_0}{2} \sum_j \sigma_j^z + \omega a^\dagger a + \frac{g}{\sqrt{N}} (a^\dagger + a) \sum_j \sigma_j^x, \tag{2}$$

where the first term describes the coupling of the spins to a magnetic field $\omega_0/2 > 0$ in $z$ direction. The second term defines the energy of the single light mode with frequency $\omega > 0$. The last term induces a coupling between light and matter degrees of freedom with a coupling constant $g/\sqrt{N} \geq 0$, with $N$ being the number of spins in the system.

We restrict ourselves to the non-superradiant phase with $\omega_0, \omega \gg g$ inducing two quasi-particle types, namely photons and magnons. The photons are given as excitations of the light field, the magnons are defined as spin-flips out of the fully polarized state induced by the magnetic field. We can rewrite the Dicke Hamiltonian with hardcore bosonic creation and annihilation operators $b_j^\dagger$, $b_j$ via the Matsubara-Matsuda transformation [37] as

$$\mathcal{H}_{\mathrm{D}} = E_0 + \omega_0 \sum_j n_j + \omega a^\dagger a + \frac{g}{\sqrt{N}} (a^\dagger + a) \sum_j (b_j^\dagger + b_j), \tag{3}$$

with $E_0 = -\omega_0 N/2$ being the ground-state energy of $\mathcal{H}_{\mathrm{D}}$ for $g = 0$ and $n_j = b_j^\dagger b_j$ the number operator.

In addition, we define a pure matter Hamiltonian of the form

$$\mathcal{H}_{\mathrm{matter}} = \sum_{j,\delta} c_\delta b_j^\dagger b_{j+\delta} + \sum_{j,\delta_1,\delta_2,\delta_3} c_{\delta_1,\delta_2,\delta_3} b_j^\dagger b_{j+\delta_1}^\dagger b_{j+\delta_2} b_{j+\delta_3}, \tag{4}$$

similarly to generalizations done before [38–41]. The first term describes (long-range) hopping processes of magnonic excitations, while the second one describes (long-range) interactions including density-density terms and correlated hopping terms. The sums over $\delta_i$ describe all possible distances on a chosen lattice. To keep the sums over all distances finite, we restrict the coefficients to stay finite in the limit of $N \to \infty$ as

$$\sum_\delta |c_\delta| < \infty, \qquad \sum_{\delta_1,\delta_2,\delta_3} |c_{\delta_1,\delta_2,\delta_3}| < \infty. \tag{5}$$

For our findings we further restrict ourselves onto the low-energy subspace demanding a finite expectation value for the number of magnonic and photonic excitations in the thermodynamic limit as

$$\left\langle \sum_i n_i \right\rangle < \infty, \qquad \langle a^\dagger a \rangle < \infty. \tag{6}$$

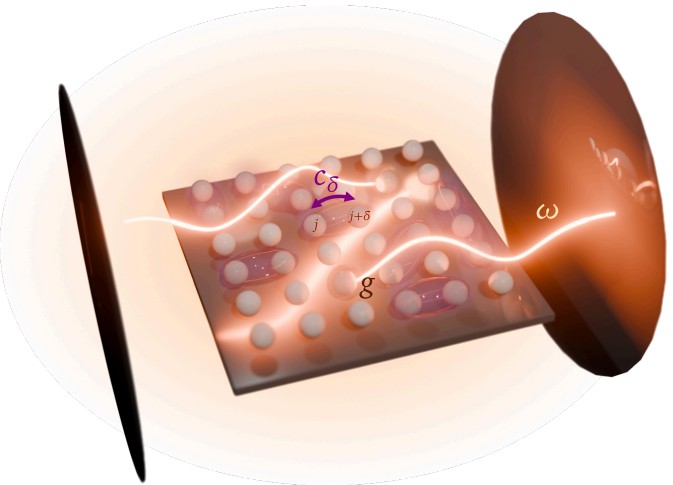

Figure 1: Visualization of the general light-matter Hamiltonian $\mathcal{H}$ from Eq. (1) on a square geometry. The model consists out of a large number of spin-1/2 particles (white spheres), with a local energy splitting caused by a uniform magnetic field, which interact with each other via (long-range) hopping and interaction terms, as introduced in Eq. (4), given in purple. The interacting matter system is put into a cavity with a single light mode of frequency $\omega$. The spins and the light mode interact via a Dicke coupling proportional to $g$.

In Fig. 1 we sketch the constituents of the model on a square lattice geometry, while we will not restrict ourselves on any particular lattice structure in the following. For parts of the discussion it is convenient to express the Hamiltonian in momentum basis via Fourier transformation. We write the operators in momentum space as

$$\tilde{b}_k = \frac{1}{\sqrt{N}} \sum_j e^{-ikj} b_j \,, \qquad \tilde{b}_k^\dagger = \frac{1}{\sqrt{N}} \sum_j e^{ikj} b_j^\dagger \,, \tag{7}$$

with $k$ being the momentum of the annihilated or created magnon. The Hamiltonian can thus be written as

$$\mathcal{H} = E_0 + \omega a^\dagger a + \left( \omega_0 + \sum_\delta c_\delta \right) \tilde{b}_0^\dagger \tilde{b}_0 + g(a^\dagger + a)(\tilde{b}_0^\dagger + \tilde{b}_0) + \sum_{k \neq 0} \left( \omega_0 + \sum_\delta c_\delta e^{ik\delta} \right) \tilde{b}_k^\dagger \tilde{b}_k$$

$$+ \frac{1}{N} \sum_{k_1,k_2,p_1,p_2} \sum_{\delta_1,\delta_2,\delta_3} c_{\delta_1,\delta_2,\delta_3} e^{i\delta_2 p_1 + i\delta_3 p_2 - i\delta_1 k_2} \tilde{b}_{k_1}^\dagger \tilde{b}_{k_2}^\dagger \tilde{b}_{p_1} \tilde{b}_{p_2} \delta_{k_1+k_2,p_1+p_2} \,, \tag{8}$$

with $\delta_{k_1+k_2,p_1+p_2}$ being the Kronecker delta preserving momentum conservation. See App. A for an in-depth derivation of Eq. (8).

For the case of vanishing $c_{\delta_1,\delta_2,\delta_3}$, we obtain again the Dicke model with a rescaled magnetic field, depending on the momentum for $c_\delta \neq 0$. In the thermodynamic limit we can show that the momentum operators fulfill the bosonic commutation relation $[\tilde{b}_p, \tilde{b}_k^\dagger] \xrightarrow{N \to \infty} \delta_{p,k}$ for the low-energy spectrum (see App. A). Thus, the Dicke model can be solved analytically by performing a Bogoliubov transformation in the $k = 0$ subspace, as described in [42]. All other $k \neq 0$ sectors are already in diagonal form. Therefore, we can read off the energies directly.

The general case $c_{\delta_1,\delta_2,\delta_3} \neq 0$ is not block diagonal in momentum space, as the matter-matter interactions induce scattering of magnon pairs, coupling different single-magnon mo-

menta. For the further discussion, we will define

$$\bar{\mathcal{H}}_{\mathrm{D}} \equiv E_0 + \omega a^{\dagger} a + \left(\omega_0 + \sum_{\delta} c_{\delta}\right) \tilde{b}_0^{\dagger} \tilde{b}_0 + g\left(a^{\dagger} + a\right)\left(\tilde{b}_0^{\dagger} + \tilde{b}_0\right), \tag{9}$$

$$\mathcal{H}_{\mathrm{MM}} \equiv \sum_{k \neq 0}\left(\omega_0 + \sum_{\delta} c_{\delta} e^{ik\delta}\right) \tilde{b}_k^{\dagger} \tilde{b}_k$$
$$+ \frac{1}{N} \sum_{k_1,k_2,p_1,p_2} \sum_{\delta_1,\delta_2,\delta_3} c_{\delta_1,\delta_2,\delta_3} e^{i\delta_2 p_1 + i\delta_3 p_2 - i\delta_1 k_2} \tilde{b}_{k_1}^{\dagger} \tilde{b}_{k_2}^{\dagger} \tilde{b}_{p_1} \tilde{b}_{p_2} \delta_{k_1+k_2,p_1+p_2}, \tag{10}$$

where $\bar{\mathcal{H}}_{\mathrm{D}}$ is the rescaled Dicke Hamiltonian from the first line of Eq. (8) and $\mathcal{H}_{\mathrm{MM}}$ all matter processes including $k \neq 0$ magnons. In the following, we will motivate and show that $\bar{\mathcal{H}}_{\mathrm{D}}$ and $\mathcal{H}_{\mathrm{MM}}$ decouple in the thermodynamic limit, given the prerequisites in Eqs. (5) and (6) to stay in the non-superradiant phase, so that we can solve the two parts of the Hamiltonian independently from each other.

## 2.2 Physical intuition

This subsection is not meant as a rigorous proof of our findings, but shall rather motivate on a physical level why the proof in Subsec. 2.3 works.

Apart from the correlated processes proportional to $c_{\delta_1,\delta_2,\delta_3}$, the general Hamiltonian $\mathcal{H}$ in Eq. (1) is solely made up of uncorrelated one-magnon terms in the matter part, including number operators and hopping processes. This means there is no interaction between magnonic excitations, apart from the hardcore constraint coming from the spin-1/2 sites. Therefore it is beneficial to transform the Hamiltonian into momentum space, as done in Eq. (8), obtaining effectively free magnon modes with different momenta. In this basis, the light-matter interaction only couples to the $k = 0$ sector. As the $\tilde{b}_k^{(\dagger)}$ operators obey bosonic commutation relations in the thermodynamic limit for the low-energy spectrum (see App. A), it is intuitive that only the $k = 0$ part of the effective hopping in $\mathcal{H}_{\mathrm{matter}}$ contributes to the Dicke dynamics, as the other magnon modes with momenta $k \neq 0$ do not couple to the light mode. Thus, the eigenstates of $\bar{\mathcal{H}}_{\mathrm{D}}$ in Eq. (9) are hybridized excitations of the $k = 0$ magnon mode and the photonic cavity mode.

Moving on to the interaction processes $\mathcal{H}_{\mathrm{MM}}$ in Eq. (10), the terms differ fundamentally to all other terms in $\mathcal{H}$ as they couple two magnons with each other. This results in effective scattering processes of the form

$$\frac{1}{N} \sum_{k_1,k_2,p_1,p_2} \sum_{\delta_1,\delta_2,\delta_3} c_{\delta_1,\delta_2,\delta_3} e^{i\delta_2 p_1 + i\delta_3 p_2 - i\delta_1 k_2} \tilde{b}_{k_1}^{\dagger} \tilde{b}_{k_2}^{\dagger} \tilde{b}_{p_1} \tilde{b}_{p_2} \delta_{k_1+k_2,p_1+p_2}, \tag{11}$$

coupling magnons with different momenta with each other. Only considering these correlated processes, we obtain eigenstates with (multiple) magnon pairs with a fixed distance to each other depending on the coefficients $c_{\delta_1,\delta_2,\delta_3}$, in contrast to the free magnon modes from the rest of the Hamiltonian. This leads to a localization of the states in real space, directly corresponding to a broadening in momentum space. Thus, the weight of the states on a single frequency is reduced, leading to a vanishing weight in the thermodynamic limit, as can be shown. As only the $k = 0$ mode couples to the light field, it is reasonable that Eq. (11) will decouple from the light part in $\mathcal{H}_{\mathrm{D}}$ for $N \to \infty$ in the low-energy subspace.

So, the low-energy sector of $\mathcal{H}$ features three kinds of dynamics in the matter part in the thermodynamic limit, which we will prove in subsection 2.3. The corresponding low-energy spectrum is sketched in Fig. 2 for a paradigmatic system. It contains

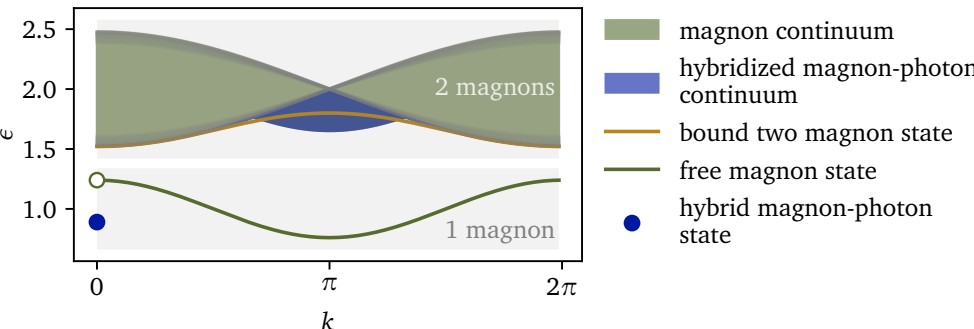

Figure 2: Sketch of the low-energy spectrum of a paradigmatic Hamiltonian $\mathcal{H}$ in Eq. (1) for small light-matter couplings. The parameters of the exemplary model are chosen appropriately to show the typical features of this class of systems. The energy of the perturbed photon states are omitted for clarity. The one-magnon subspace consists out of a conventional free mode (green line) with an energy jump at $k = 0$, where the magnon mode couples to the light mode (blue dot). The two-magnon subspace consists out of a conventional continuum (green area) complemented by composite two-magnon states with one hybridized magnon at $k = 0$ and a magnon with $k \neq 0$ that does not couple to the light mode (blue area). For suitable $\mathcal{H}_{\mathrm{MM}}$ one or more bound states can form (orange line) that are not affected by the light-matter coupling.

1. a free magnon state with momentum $k = 0$ forming a hybridized state with photons from the cavity due to the light-matter coupling term in $\bar{\mathcal{H}}_{\mathrm{D}}$ (blue dot),

2. free magnon states with non-vanishing momentum in $\mathcal{H}_{\mathrm{MM}}$ that do not couple to the light mode (green line),

3. bound magnon-pair states with a fixed distance in $\mathcal{H}_{\mathrm{MM}}$ that do not couple to the light mode (orange line).

While the first one commutes with the latter two, the second and third part may only be described together, as it is common for correlated matter Hamiltonians. The first part can be solved analogously to the Dicke model by applying a Bogoliubov transformation because this part does not involve any matter-matter interactions (see App. C). As $\mathcal{H}_{\mathrm{MM}}$ conserves the number of magnons, this part of the model is block diagonal with respect to the number of particles and can be solved, e.g., with exact diagonalization.

## 2.3 General case

To show on a general level that $\bar{\mathcal{H}}_{\mathrm{D}}$ and $\mathcal{H}_{\mathrm{MM}}$ from Eqs. (9) and (10) decouple, we prove in this concluding derivation section that the commutator $[\bar{\mathcal{H}}_{\mathrm{D}}, \mathcal{H}_{\mathrm{MM}}]$ vanishes in the thermodynamic limit for the given prerequisites. We therefore perform calculations on finite systems with $N$ sites and afterwards take the limit $N \to \infty$. Contrary to intuition, the terms $\bar{\mathcal{H}}_{\mathrm{D}}$ and $\mathcal{H}_{\mathrm{MM}}$ with different distinct momenta $p \neq k$ do not commute for finite systems as $[\tilde{b}_p, \tilde{b}_k^\dagger] \neq 0$ (see App. A for details). Thus, we have to include these terms in the calculation to rigorously show that the commutator vanishes in the thermodynamic limit. In the following we split the calculation of the commutator into the two terms of $\mathcal{H}_{\mathrm{MM}}$, starting with the free-particle term. First, we

find

$$\left[\bar{\mathcal{H}}_{\mathrm{D}}, \sum_{k\neq 0}\left(\omega_0 + \sum_\delta c_\delta e^{ik\delta}\right)\tilde{b}_k^\dagger \tilde{b}_k\right] = \sum_{k\neq 0}\left(\omega_0 + \sum_\delta c_\delta e^{ik\delta}\right) \tag{12}$$
$$\times\left(\left(\omega_0 + \sum_\delta c_\delta\right)[\tilde{b}_0^\dagger \tilde{b}_0, \tilde{b}_k^\dagger \tilde{b}_k] + g(a^\dagger + a)[\tilde{b}_0^\dagger + \tilde{b}_0, \tilde{b}_k^\dagger \tilde{b}_k]\right).$$

We can express both commutators in Eq. (12) in terms of the commutators

$$\left[\tilde{b}_0, \tilde{b}_k^\dagger \tilde{b}_k\right] \quad \text{and} \quad \left[\tilde{b}_0^\dagger, \tilde{b}_k^\dagger \tilde{b}_k\right], \tag{13}$$

using the product identity $[AB,C] = [A,C]B + A[B,C]$. In App. B we show that the norms of

$$\sum_{k\neq 0} e^{ik\delta}\left[\tilde{b}_0, \tilde{b}_k^\dagger \tilde{b}_k\right] \quad \text{and} \quad \sum_{k\neq 0} e^{ik\delta}\left[\tilde{b}_0^\dagger, \tilde{b}_k^\dagger \tilde{b}_k\right], \tag{14}$$

scale with $N^{-1/2}$ and thus are zero in the thermodynamic limit. As the sum over all $|c_\delta|$ has a finite value and $\langle a^\dagger + a\rangle, \langle \tilde{b}_k\rangle, \langle \tilde{b}_k^\dagger\rangle < \infty$ for the low-energy subspace, we conclude that the commutator of Eq. (12) vanishes in the thermodynamic limit.

For the correlated hopping term in $\mathcal{H}_{\mathrm{MM}}$ we build the commutator analogously – using a mixed representation of momentum and real-space operators to keep notation short – as

$$[\bar{\mathcal{H}}_{\mathrm{D}}, \sum_{j,\delta_1,\delta_2,\delta_3} c_{\delta_1,\delta_2,\delta_3} b_j^\dagger b_{j+\delta_1}^\dagger b_{j+\delta_2} b_{j+\delta_3}] = \sum_{j,\delta_1,\delta_2,\delta_3} c_{\delta_1,\delta_2,\delta_3} \tag{15}$$
$$\times\left[\left(\omega_0 + \sum_\delta c_\delta\right)[\tilde{b}_0^\dagger \tilde{b}_0, b_j^\dagger b_{j+\delta_1}^\dagger b_{j+\delta_2} b_{j+\delta_3}] + g(a^\dagger + a)[\tilde{b}_0^\dagger + \tilde{b}_0, b_j^\dagger b_{j+\delta_1}^\dagger b_{j+\delta_2} b_{j+\delta_3}]\right].$$

Again, we reexpress the first commutator of Eq. (15) by the second one by using the product identity. In App. B we show that the norm of the commutators

$$\sum_j \left[\tilde{b}_0, b_j^\dagger b_{j+\delta_1}^\dagger b_{j+\delta_2} b_{j+\delta_3}\right] \quad \text{and} \quad \sum_j \left[\tilde{b}_0^\dagger, b_j^\dagger b_{j+\delta_1}^\dagger b_{j+\delta_2} b_{j+\delta_3}\right], \tag{16}$$

scale with $N^{-1/2}$. With Eq. (5) and $\langle a^\dagger + a\rangle, \langle \tilde{b}_k\rangle, \langle \tilde{b}_k^\dagger\rangle < \infty$ we conclude analogous to above that the commutator of Eq. (15) approaches zero in the thermodynamic limit for the low-energy spectrum.

Thus, the commutator $[\bar{\mathcal{H}}_{\mathrm{D}}, \mathcal{H}_{\mathrm{MM}}]$ vanishes in the thermodynamic limit. Therefore, we can discuss the rescaled Dicke model $\bar{\mathcal{H}}_{\mathrm{D}}$ and the correlated processes from $\mathcal{H}_{\mathrm{MM}}$, separately. This induces that the potential bound states evolving out of $\mathcal{H}_{\mathrm{MM}}$ do not couple to the light mode in the thermodynamic limit, agreeing on the physical intuition gained in Subsec. 2.2.

## 3 Application onto the Dicke-Ising model

After motivating and proving the main finding of this work that $\bar{\mathcal{H}}_{\mathrm{D}}$ and $\mathcal{H}_{\mathrm{MM}}$ commute in the thermodynamic limit, we will now apply it onto a concrete model, namely the Dicke-Ising model. First, we introduce the model and give an overview of the research done on it in the last years. Afterwards, we bring it into the form of Eq. (1), followed by a discussion of the two non-superradiant phases of the model, which obey the prerequisites in Eq. (6), namely the paramagnetic normal phase and the antiferromagnetic normal phase. In the end, we will compare our results, which were derived in the thermodynamic limit, with calculations on finite systems to check for the convergence towards the effective model for larger system sizes.

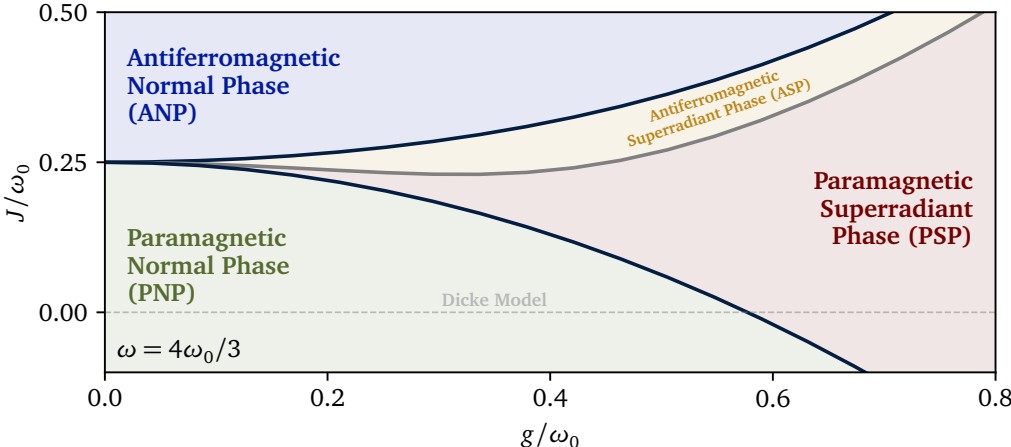

Figure 3: Phase diagram of the Dicke-Ising model for the fixed representative ratio $\omega = 4\omega_0/3$ in 1D ($c = 2$). The four phases PNP, ANP, ASP, and PSP are given in different colors. The phase transitions obtained by mean-field theory in [2] are plotted in solid lines. For the PNP and ANP we discuss effective models for the low-energy spectrum becoming exact in the thermodynamic limit. The phase transitions out of the two normal phases coincide with the closing of lowest gap in the presented effective theory, denoted with black solid lines. For $J = 0$, we recover the standard Dicke model, indicated with a faint dashed line.

## 3.1 Introduction to the model

The Dicke-Ising model is defined as a combination of the Dicke model with an additional Ising interaction in the direction of the external magnetic field. It can be written as [2]

$$\mathcal{H}_{\mathrm{DI}} = \frac{\omega_0}{2} \sum_j \sigma_j^z + \omega a^\dagger a + \frac{g}{\sqrt{N}}(a^\dagger + a) \sum_j \sigma_j^x + J \sum_{\langle j,l \rangle} \sigma_j^z \sigma_l^z, \tag{17}$$

with ferromagnetic (antiferromagnetic) Ising interactions for $J < 0$ ($J > 0$) and $N$ being the number of sites. Due to the competition of the Ising interactions with the magnon-photon coupling, the model features a variety of phases. In [2] the phase diagram for antiferromagnetic Ising interactions was investigated with mean-field theory and a classical approximation of the matter part. This yields four distinct phases, as plotted in Fig. 3. According to this mean-field treatment, all quantum phase transitions between the four phases are of second order. In contrast, recent works using perturbation theory and exact diagonalization [1] showed that the phase transition at the limiting case $\omega_0 = 0$ is of first order in 1D. A study on the full Dicke-Ising model using quantum Monte Carlo [36] has found that the first-order transition survives around the limit $\omega_0 = 0$ in 1D and 2D. For larger $\omega_0$ the gap shrinks at the phase-transition point until changing to a continuous phase transition, as predicted by the mean-field study. We refer to [36] for a quantitative discussion of the nature of the phase transitions.

The different phases can be characterized by the photon density $\langle a^\dagger a \rangle / N$ and the $x$-magnetization of the spins. For small $g$, the 'paramagnetic normal phase' (PNP) and the 'antiferromagnetic normal phase' (ANP) have zero photon density. For larger $g$, the remaining two phases, namely the 'paramagnetic superradiant phase' (PSP) and the intermediate 'antiferromagnetic superradiant phase' (ASP), have a non-vanishing photon density. The photon density is therefore the relevant order parameter. Because of our prerequisites in Eq. (6), we concentrate on the former two phases, staying adiabatically connected to the case of no light-matter interaction $g = 0$. We discuss our findings for the two phases in separate Subsecs. 3.2 and 3.3, as the derivation of Eq. (1) from Eq. (17) depends on the phase. It turns out that the

phase transitions out of the PNP and ANP obtained by mean-field calculations coincide with the closing of the lowest excitation level in the effective theory that we derived (as plotted in Fig. 3).

For $g = 0$, the Dicke-Ising Hamiltonian boils down to a longitudinal-field Ising model

$$\mathcal{H}_\mathrm{I} = \frac{\omega_0}{2} \sum_j \sigma_j^z + J \sum_{\langle j,l \rangle} \sigma_j^z \sigma_l^z \,, \tag{18}$$

ignoring the light part of the Hamiltonian, as it behaves trivially in this limit. The phase transition between the paramagnetic and antiferromagnetic phase in this limit is of first order. The critical point $J_\mathrm{crit}$ depends on the connectivity of the chosen lattice, yielding

$$J_\mathrm{crit} = \frac{\omega_0}{2c} \,, \tag{19}$$

on a bipartite lattice, with $c$ being the number of bonds per site, e.g., $c = 2$ for a chain geometry.

For the limiting case $\omega_0 = 0$, one recovers the case studied in [1] for 1D with the Hamiltonian

$$\mathcal{H}_\mathrm{DI} = \omega a^\dagger a + \frac{g}{\sqrt{N}}(a^\dagger + a) \sum_j \sigma_j^x + J \sum_{\langle j,l \rangle} \sigma_j^z \sigma_l^z \,. \tag{20}$$

In the work a first-order phase transition between the normal and superradiant phase was found at

$$J_\mathrm{crit} \approx 0.2988 \frac{g^2}{\omega} \,, \tag{21}$$

by comparing the ground-state energies of the two phases [1, 36]. In the weak light-matter coupling phases, the ground-state energy per site stays unaffected by perturbations in $g$ [1], thus having the constant energy

$$\epsilon_\mathrm{PNP} = \frac{E_\mathrm{PNP}}{N} = -\frac{\omega_0}{2} + \frac{c}{2}J \,, \qquad \epsilon_\mathrm{ANP} = \frac{E_\mathrm{ANP}}{N} = -\frac{c}{2}J \,, \tag{22}$$

for PNP and ANP on bipartite lattices, respectively. In contrast, the excitations are affected by the light-matter coupling in these low-coupling phases, as we discuss in the following.

## 3.2 Paramagnetic normal phase

The paramagnetic normal phase is adiabatically connected to the limit $J = g = 0$ so that photons and spin flips are elementary excitations in this phase. Thus, it is suitable to introduce hardcore bosons in the same way as done in Eq. (3), resulting in the Hamiltonian

$$\mathcal{H}_\mathrm{DI} = E_0 + (\omega_0 - 2cJ) \sum_j n_j + \omega a^\dagger a + \frac{g}{\sqrt{N}}(a^\dagger + a) \sum_j (b_j^\dagger + b_j) + 4J \sum_{\langle j,l \rangle} n_j n_l \,, \tag{23}$$

with $E_0 = -\omega_0 N/2 + JcN/2$ and $c$ being the number of bonds per site. We can now map $\mathcal{H}_\mathrm{DI}$ to Eq. (1) up to a constant offset by defining

$$c_\delta \equiv -2cJ \, \delta_{\delta,0} \,, \qquad c_{\delta_1,\delta_2,\delta_3} \equiv 4J \, \delta_{|\delta_1|,1} \delta_{\delta_2,0} \delta_{\delta_1,\delta_3} \,, \tag{24}$$

with $\delta_{\cdot,\cdot}$ being the Kronecker delta, effectively setting all distances in Eq. (4) to zero or nearest neighbors (by defining a distance $\delta_i$ with $|\delta_i| = 1$ as nearest neighbor) and fulfilling the prerequisites of Eq. (5).

As the low-energy subspace fulfills Eq. (6), we can apply the finding of the last section onto this model. The rescaled Dicke Hamiltonian in momentum space from Eq. (9) has the form

$$\bar{\mathcal{H}}_\mathrm{D} = E_0 + \omega a^\dagger a + (\omega_0 - 2cJ) \, \tilde{b}_0^\dagger \tilde{b}_0 + g \left( a^\dagger + a \right) \left( \tilde{b}_0^\dagger + \tilde{b}_0 \right) \,, \tag{25}$$

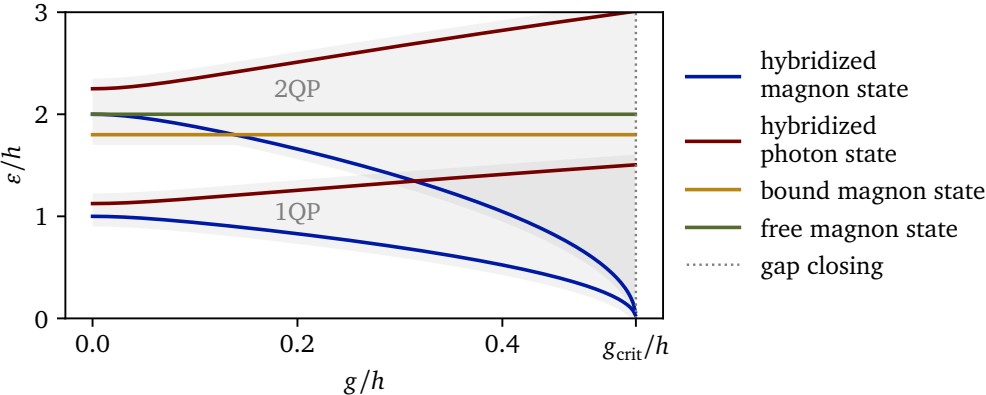

Figure 4: Low-energy spectrum of the Dicke-Ising model in the paramagnetic normal phase. The parameters are chosen as $\omega/h = 9/8$, $J/h = -1/20$, and $c = 2$ (chain geometry) with $h \equiv \omega_0 - 2cJ$ at vanishing momentum $k = 0$. For the spectrum only dressed states with one or two quasiparticles are considered, omitting the mixed 2QP states with one dressed magnon and photon for clarity of the plot. The unperturbed magnon (photon) states hybridize for finite $g$ leading to a decreased (increased) energy. At $g_{\text{crit}}$ the magnon gap closes, indicating the end of validity of the found eigenvalues. The free magnons and bound magnons in the 2QP sector are not coupled to the light mode and therefore stay constant in energy when varying $g$.

and can be diagonalized in the thermodynamic limit by using a Bogoliubov transformation (see App. C for the derivation). The diagonalized Hamiltonian can be written as

$$\bar{\mathcal{H}}_{\text{D}} = E_0 + \varepsilon_+ b_+^\dagger b_+ + \varepsilon_- b_-^\dagger b_- \,, \tag{26}$$

with $b_+^{(\dagger)}$, $b_-^{(\dagger)}$ being a linear combination of the $a^{(\dagger)}$, $\tilde{b}_0^{(\dagger)}$ operators and the corresponding eigenenergies

$$\varepsilon_\pm = \sqrt{\frac{\omega^2}{2} + \frac{h^2}{2} \pm \frac{\sqrt{(\omega^2 - h^2)^2 + 16\omega g^2 h}}{2}} \,, \tag{27}$$

defining $h \equiv \omega_0 - 2cJ$ as the coefficient of the rescaled magnetic-field term. The remaining part of the Hamiltonian does not couple to the light field and thus stays unaffected by varying $g$. We can write $\mathcal{H}_{\text{MM}}$ in the original form with a correction term for $k = 0$ as

$$\mathcal{H}_{\text{MM}} = (\omega_0 - 2cJ)\sum_j n_j + 4J \sum_{\langle j,l \rangle} n_j n_l - (\omega_0 - 2cJ)\tilde{b}_0^\dagger \tilde{b}_0 \,. \tag{28}$$

Note that $\mathcal{H}_{\text{MM}}$ is no longer diagonal in the real-space basis due to the correction term. Nonetheless, the interaction term and the correction term commute in the thermodynamic limit for the low-energy subspace, as can be shown analogously to Subsec. 2.3. So, we obtain a flat dispersion for $k \neq 0$ in the one-magnon sector with energy $\varepsilon_{k \neq 0} = \omega_0 - 2cJ$. The higher particle channels involve (anti-)bound states for neighboring magnons due to the interaction term in $\mathcal{H}_{\text{MM}}$. The energy of a free magnon pair is given directly as twice the one-magnon energy, as can be found using the free particle approximation [43]. The (anti-)bound magnon-pair state is given by two magnons next to each other to (maximize) minimize the Ising term in Eq. (28). In the thermodynamic limit the energy of the (anti-)bound state is given exactly as $\varepsilon_{\text{bound}} = 2 \cdot (\omega_0 - 2cJ) + 4J$, taking twice the free magnon energy plus the energy offset caused by the nearest-neighbor Ising interaction.

In Fig. 4 we plot the excitation energies for $k = 0$ varying the light-matter interaction $g$. For the plot, we have fixed the parameters of the model to be $\omega/h = 9/8$, $J/h = -1/20$ with

$h \equiv \omega_0 - 2cJ$ and a connectivity of $c = 2$, as present for a chain geometry. Note that other values for the parameters give qualitatively similar results. To keep the visualization light, we only plot the energies of states with up to two quasiparticles. For the unperturbed case ($g = 0$), we have the one-magnon (in blue), one-photon (in red), bound two-magnon (in orange), free two-magnon (in green and blue), and two-photon state (in red) from lower to higher excitation energies. As argued, the bound magnon state and the free magnon states, where all individual magnons have non-vanishing momenta, stay constant in energy as they do not couple to the light mode (in orange and green). The energies of the two-magnon and two-photon states are given by the doubled value of the respective one-particle energies, as they form bosonic modes in the thermodynamic limit. For the critical value

$$g_{\text{crit}} = \sqrt{h\omega}/2, \tag{29}$$

the gap of the magnon mode closes with a critical exponent of $1/2$ indicating a second-order phase transition. This critical value coincides with the one calculated by [2] with mean-field methods. Nonetheless, it does not rule out a potential first-order phase transition for smaller $g$ values, as we do not have any information about the energetics of the superradiant phase. Therefore, the first-order nature of phase transition lines around $\omega_0 = 0$ observed by perturbation theory, exact diagonalization [1] and quantum Monte Carlo calculations [36] is in no contradiction to our findings. As can be seen in Eq. (27) whether the renormalized one-magnon or one-photon gap closes depends on which of the two have a higher unperturbed energy. For deeper insights of the excited states in low-energy space, like corresponding observables, we can use the eigenvectors obtained by the Bogoliubov transformation, as presented in App. C.

## 3.3  Antiferromagnetic normal phase

For the antiferromagnetic normal phase we will restrict ourselves to bipartite lattices (denoting $A, B$ as the two subsets of sites) to avoid any kind of geometric frustration caused by the Ising interaction for $J > 0$. For models with geometric frustration, a straightforward mapping to the generalized Dicke model in Eq. (1) is not possible, as the model features an extensive ground-state degeneracy in the Ising limit, which is not contained in our model. The phase is adiabatically connected to the case of $\omega_0 = g = 0$, with an unperturbed ground state with alternating spin orientations and an empty cavity mode. To bring the Hamiltonian closer to the form of Eq. (1), we first have to apply a rotation on one of the sublattices (here sublattice $A$). Therefore, we apply a sublattice rotation $U$ about the $x$-axis giving

$$\sigma_j^z \mapsto -\sigma_j^z, \qquad \sigma_j^x \mapsto \sigma_j^x, \tag{30}$$

for $j \in A$, altering the eigenfunctions but not the eigenvalues of the Hamiltonian. The rotated Dicke-Ising Hamiltonian

$$\mathcal{H}_{\text{RDI}} \equiv U\mathcal{H}_{\text{DI}}U^\dagger = \frac{\omega_0}{2}\sum_j (-1)^j \sigma_j^z + \omega a^\dagger a + \frac{g}{\sqrt{N}}(a^\dagger + a)\sum_j \sigma_j^x - J\sum_{\langle j,l \rangle} \sigma_j^z \sigma_l^z, \tag{31}$$

possesses a fully polarized ground state for $\omega_0 = g = 0$, as the sign in front of the Ising interaction is flipped by the transformation. The expression $(-1)^j$ is the shorthand notation for

$$(-1)^j := \begin{cases} -1, & \text{if } j \in A, \\ +1, & \text{else}, \end{cases} \tag{32}$$

inspired by the usual notation in the case of a chain geometry. We perform a mapping to hardcore bosons, as done in Eq. (3), resulting in

$$\mathcal{H}_{\text{RDI}} = E_0 + \sum_j \left((-1)^j \omega_0 + 2cJ\right)n_j + \omega a^\dagger a + \frac{g}{\sqrt{N}}(a^\dagger + a)\sum_j (b_j^\dagger + b_j) - 4J\sum_{\langle j,l \rangle} n_j n_l, \tag{33}$$

with $E_0 = -JcN/2$. While we can map parts of $\mathcal{H}_{\mathrm{RDI}}$ to $\mathcal{H}_{\mathrm{matter}}$ in Eq. (4) up to a constant offset with

$$c_\delta \equiv 2cJ \cdot \delta_{\delta,0}, \qquad c_{\delta_1,\delta_2,\delta_3} \equiv -4J \cdot \delta_{|\delta_1|,1}\delta_{\delta_2,0}\delta_{\delta_1,\delta_3}, \tag{34}$$

the Dicke Hamiltonian $\mathcal{H}_{\mathrm{D}}$ can not directly be recovered due to the staggered magnetic field in $\mathcal{H}_{\mathrm{RDI}}$. Instead, we define two separate magnon modes in momentum space for the $A$ and $B$ sublattice of the form

$$\tilde{b}_{k,s} = \frac{1}{\sqrt{N/2}}\sum_{j\in s}e^{-\mathrm{i}kj}b_j, \qquad \tilde{b}^\dagger_{k,s} = \frac{1}{\sqrt{N/2}}\sum_{j\in s}e^{\mathrm{i}kj}b^\dagger_j, \tag{35}$$

with $s,s' \in \{A,B\}$. The new operators fulfill pairwise bosonic commutation relations in the low-energy sector as $[\tilde{b}_{p,s}, \tilde{b}^\dagger_{k,s'}] \xrightarrow{N\to\infty} \delta_{p,k}\delta_{s,s'}$. With this we rewrite $\mathcal{H}_{\mathrm{RDI}}$ as

$$\begin{aligned}
\mathcal{H}_{\mathrm{RDI}} = {}& E_0 + \omega a^\dagger a + (2cJ-\omega_0)\sum_k \tilde{b}^\dagger_{k,A}\tilde{b}_{k,A} + (2cJ+\omega_0)\sum_k \tilde{b}^\dagger_{k,B}\tilde{b}_{k,B} \\
& + \frac{g}{\sqrt{2}}(a^\dagger + a)\left(\tilde{b}^\dagger_{0,A} + \tilde{b}^\dagger_{0,B} + \tilde{b}_{0,A} + \tilde{b}_{0,B}\right) \\
& - \frac{8J}{N}\sum_{k_1,k_2,p_1,p_2}\sum_{|\delta|=1}e^{\mathrm{i}\delta(p_2-k_2)}\tilde{b}^\dagger_{k_1,A}\tilde{b}^\dagger_{k_2,B}\tilde{b}_{p_1,A}\tilde{b}_{p_2,B}\delta_{k_1+k_2,p_1+p_2}.
\end{aligned} \tag{36}$$

The form is analogous to the general case in Eq. (1), with the difference that there are two bosonic modes which are coupled via the light mode and the Ising interaction. We can prove analogously to before that the Ising term and the $k \neq 0$ modes decouple from the $k = 0$ magnon modes which are coupled to the light field for the prerequisites in Eq. (6) in the thermodynamic limit.

Thus, we have again a simplified Hamiltonian at $k = 0$ describing the complete dynamics induced by the light field, analogous to the renormalized Dicke model $\bar{\mathcal{H}}_{\mathrm{D}}$, reading

$$\begin{aligned}
\bar{\mathcal{H}}_{\mathrm{RD}} = {}& E_0 + \omega a^\dagger a + (2cJ-\omega_0)\tilde{b}^\dagger_{0,A}\tilde{b}_{0,A} + (2cJ+\omega_0)\tilde{b}^\dagger_{0,B}\tilde{b}_{0,B} \\
& + \frac{g}{\sqrt{2}}(a^\dagger + a)\left(\tilde{b}^\dagger_{0,A} + \tilde{b}^\dagger_{0,B} + \tilde{b}_{0,A} + \tilde{b}_{0,B}\right).
\end{aligned} \tag{37}$$

We can solve this system again with a Bogoliubov transformation, as shown in App. C, yielding the Hamiltonian

$$\bar{\mathcal{H}}_{\mathrm{RD}} = E_0 + \epsilon_+ b^\dagger_+ b_+ + \epsilon_- b^\dagger_- b_- + \epsilon_\star b^\dagger_\star b_\star, \tag{38}$$

with three elementary excitation types, which are built out of the photon mode and the two magnon modes for the even and odd sites. While the eigenenergies $\epsilon_+$, $\epsilon_-$, $\epsilon_\star$ can be calculated analytically, we omit them here due to complexity of the analytic expressions (see the supplementary data for the full expressions) [44].

For the rest of $\mathcal{H}_{\mathrm{RDI}}$ we can argue, analogous to the paramagnetic phase with $\mathcal{H}_{\mathrm{MM}}$, that all non-diagonal corrections for $k = 0$ will commute with the rest of the matter-matter interactions. So, we again obtain flat modes for $k \neq 0$ in the one-magnon sector and bound states in the higher particle channels.

We plot the low-energy spectrum for vanishing momentum varying the light-matter interaction in Fig. 5. We define $h \equiv 2cJ - \omega_0$ as the lowest unperturbed excitation energy and fix the parameters to representative values $\omega/h = 9/8$, $(2cJ + \omega_0)/h = 13/8$, and $c = 2$. As before, we only plot energies of states with up to two quasiparticles. The spectrum offers a richer structure than for the paradigmatic phase in Fig. 4 as we have two independent dressed magnon modes (both plotted in blue). This also results in three flat free-magnon bands in the 2QP sector (in green), coming from the combinations of the two elementary magnon modes.

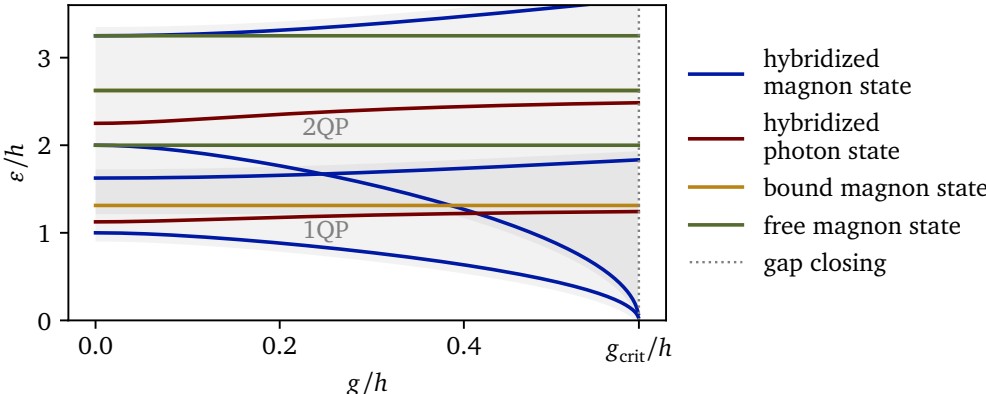

Figure 5: Low-energy spectrum of the Dicke-Ising model in the antiferromagnetic normal phase. The parameters are chosen as $\omega/h = 9/8$, $(2cJ + \omega_0)/h = 13/8$, and $c = 2$ (chain geometry) with $h \equiv 2cJ - \omega_0$ at vanishing momentum $k = 0$. For the spectrum only dressed states with one or two quasiparticles are considered, omitting the mixed 2QP states with one dressed magnon and photon for clarity of the plot. The three unperturbed magnon (photon) states hybridize for finite $g$ leading to a decreased (increased) energy. At $g_{\mathrm{crit}}$, one of the elementary magnon gaps closes, indicating the end of validity of the found eigenvalues. The free magnons and bound magnons in the 2QP sector are not coupled to the light mode and therefore stay constant in energy when varying $g$.

Using the free particle approximation, the energy of the flat bands is given as the sum of the energies of the 1QP modes. Note that the 1QP and 2QP sectors overlap for all values of $g$, as the bound magnon state (in orange), with two excitations next to each other, has a lower energy than the upper 1QP magnon band. In the thermodynamic limit the energy of the bound state is given as

$$\varepsilon_{\mathrm{bound}} = (2cJ + \omega_0) + (2cJ - \omega_0) - 4J = 4J(c - 1), \tag{39}$$

being again the sum of the individual magnons (one from sublattice $A$, one from sublattice $B$) plus the energy correction from the Ising interaction. The lower magnon mode closes for the critical value

$$g_{\mathrm{crit}} = \sqrt{\omega}\sqrt{J - \frac{\omega_0^2}{16J}}, \tag{40}$$

again coinciding with the one found within mean field [2]. We are again not capable of detecting the first-order phase transition found with numerical methods [1,36] around the limit $\omega_0 = 0$, as expected.

## 3.4 Comparison to results on finite systems

After presenting results for the Dicke-Ising model using the effective model in the thermodynamic limit, in this subsection we will compare the effective model with calculations on finite systems checking for the convergence towards the effective model when increasing the system size. We perform calculations using two methods: the first being exact diagonalization (ED) and the second being the perturbative series expansion method pcst$^{++}$ [3], treating the light-matter interaction $g$ as the perturbation parameter. To keep the discussion concise, we will limit ourselves to the chain geometry and the paramagnetic normal phase using the same parameters as in 3.2, while the same is possible for other geometries and the antiferromagnetic normal phase.

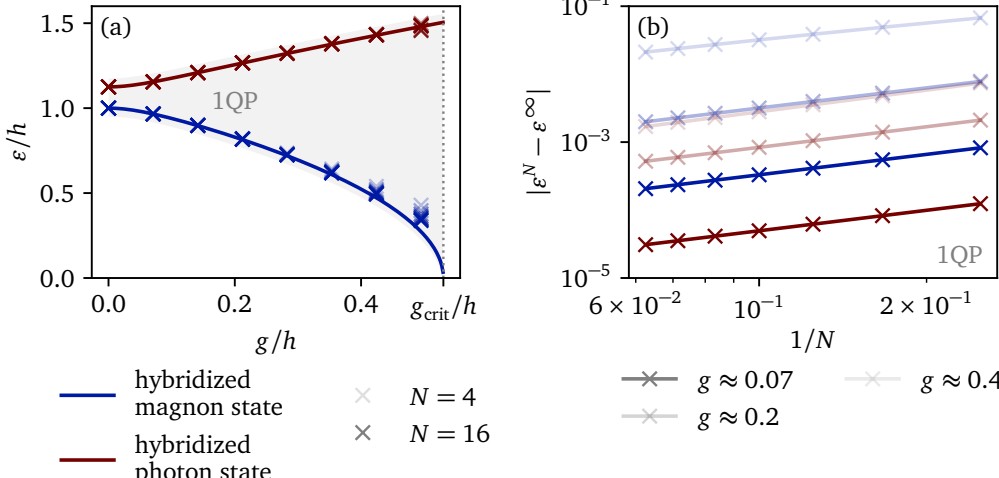

Figure 6: Comparison of the results for the 1QP sector obtained by the effective theory in the thermodynamic limit and exact diagonalization of finite systems. The parameters of the model are the same as in Fig. 4. In (a), the finite-system results are plotted as crosses for various system sizes $4 \leq N \leq 14$, the results from the effective model are given as lines. In (b), the absolute energy difference between the effective model $\varepsilon^{\infty}$ and the finite systems $\varepsilon^{N}$ is plotted for varying system sizes $N$. The different lines correspond to various $g$ values and the magnon and photon mode.

**Exact diagonalization** In Fig. 6 we compare our derived effective model with results on finite systems using ED. For that we look at the 1QP sector, namely the dressed one-magnon state (in blue) and the dressed one-photon state (in red). To determine the corresponding states in the ED calculations, we check the weight of the unperturbed 1QP states in the obtained eigenstates to choose the eigenstate which is closest to the desired unperturbed state. In Fig. 6(a) we see quite good agreement of the finite and the infinite systems for small light-matter couplings, while for couplings closer to the phase transition we obtain larger deviations. This is to be expected as the light-matter coupling, acting globally on the system, becomes non-negligible and therefore adds a stronger $N$ dependence. Additionally, the ED results can be affected by the nearby phase transition, potentially happening before the closing of the gap by a first-order transition.

In Fig. 6(b) the absolute energy difference $|\varepsilon^{N} - \varepsilon^{\infty}|$ between the effective theory $\varepsilon^{\infty}$ and the different finite systems $\varepsilon^{N}$ is plotted for various $g$ values. All differences show the same overall exponent with respect to $1/N$. Applying a fit to the data, we obtain an extrapolation of the form

$$|\varepsilon^{N} - \varepsilon^{\infty}| \propto N^{\alpha}, \quad \alpha \approx -0.99 \pm 0.02, \tag{41}$$

within the PNP for $g/h < 0.35$. In contrast, taking a point close to $g_{\text{crit}}$, we obtain no clear exponent anymore (not shown). These deviations hint for a first-order phase transition, affecting the ED results in a way that is not taken into account in the effective model thus leading to a non-vanishing difference when taking the thermodynamic limit.

We also calculate the operator norm of the commutator $[\bar{\mathcal{H}}_{\text{D}}, \mathcal{H}_{\text{MM}}]$, as defined in App. B, for the low-energy subspace of the finite systems (not shown). We find that the commutator norm clearly scales with $N^{-1/2}$, which fits perfectly to the derived scaling relation in Sec. 2. While we have no direct handle to the deviation of the energies of the finite systems in relation to the derived effective theory, we can motivate the scaling of $N^{-1}$ in Eq. (41) with perturba-

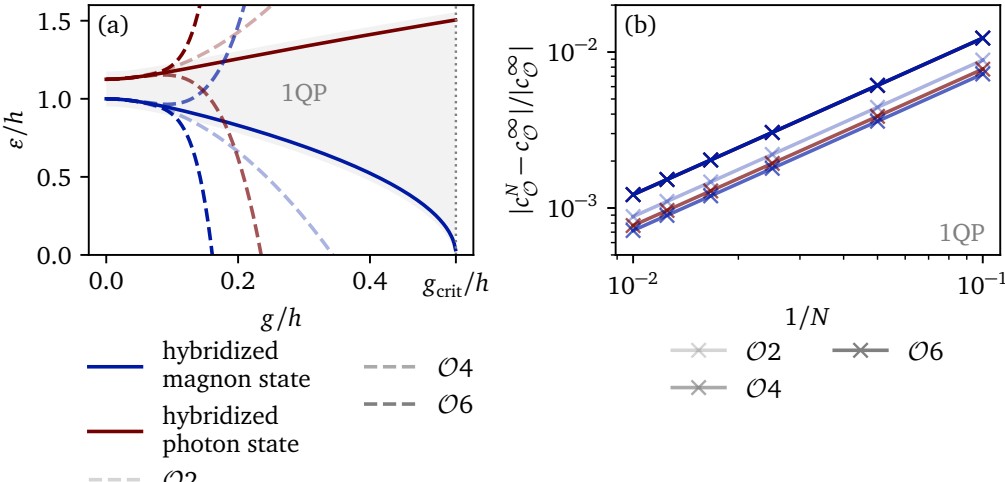

Figure 7: Comparison of the results for the 1QP sector obtained by the effective theory in the thermodynamic limit and pcst$^{++}$ on finite systems. The parameters of the model are the same as in Fig. 4. In (a), the finite-system results are plotted as dashed lines for different orders, the results from the effective model are given as solid lines. In (b), the series expansion from pcst$^{++}$ is compared to the series expansion of the effective model in the thermodynamic limit. The relative difference of the coefficients of the orders is plotted vor varying system sizes, $c_{\mathcal{O}}^N$ denoting the coefficient of order $\mathcal{O}$ of the series for the system with $N$ sites.

tion theory. As the two terms $\bar{\mathcal{H}}_D$, $\mathcal{H}_{MM}$ have a non-vanishing commutator, treating one as a perturbation to the other, yields a first potentially non-vanishing contribution in second order. As the commutator scales with $N^{-1/2}$ – both shown in the analytic derivation and ED – we obtain the squared factor of $N^{-1}$ in second order. This explains the found scaling of the energy in Eq. (41). It would be interesting to gain a deeper understanding between the interconnection of the energy and commutator scaling with $N$, which could be a future task.

**pcst$^{++}$** In Fig. 7 we compare our effective model with results on finite systems using the perturbative series expansion method pcst$^{++}$ that was introduced in [3]. The pcst$^{++}$ is a generalization of the perturbative continuous unitary transformation (pCUT) method [45, 46], often used to treat quantum many-body systems with high-order linked-cluster expansions. In contrast to pCUT, pcst$^{++}$ is capable of having an arbitrary number of quasiparticle-types in the unperturbed part of the Hamiltonian, enabling us to apply it to the Dicke-Ising model, with $g$ as the perturbation parameter. For more information about the method we refer to [3].

In Fig. 7(a) we plot the obtained series expansions in dashed lines, cutting the series at different orders, going from $\mathcal{O}2$ to $\mathcal{O}6$ for fixed $N = 100$. For both 1QP energies, the results match with the effective model up to around $g/h \approx 0.1$ before diverging. This divergence, as explained in more detail in [3], comes from the energetic vicinity of the magnon and photon mode. Due to the perturbative treatment of the model, the series expansion's convergence radius depends on the energy differences between the individual unperturbed eigenenergies in the bare Hamiltonian giving rise to the obtained divergence.

While this issue can be overcome by modifying the underlying transformation, it is not needed for the following discussion. Instead, we calculate the series expansion of the effective model's energy around $g = 0$ and compare the coefficients of the individual orders with those of the series obtained by pcst$^{++}$. The result is shown in Fig. 7(b). We plot the normalized total difference of the individual coefficients $|c_{\mathcal{O}}^N - c_{\mathcal{O}}^\infty|/|c_{\mathcal{O}}^\infty|$, where $c_{\mathcal{O}}^N$ is the coefficient of order

$\mathcal{O}$ for a system of $N$ sites obtained by pcst$^{++}$ and $c_{\mathcal{O}}^{\infty}$ is the respective coefficient of the series obtained by the effective theory. The absolute error obeys again an extrapolation of the form

$$|c_{\mathcal{O}}^N - c_{\mathcal{O}}^{\infty}| \propto N^{\alpha}, \quad \alpha \approx -1.00 \pm 0.01, \tag{42}$$

for the given coefficients obtained by pcst$^{++}$. It should be mentioned that the $c_2^N$ for the one-photon mode already matches up to computer precision with $c_2^{\infty}$ and is therefore not shown in Fig. 7(b). This can be explained by the fact that there are no finite-size corrections in second order for the energy of the one-photon state, as there are no magnons in the finite system that would alter the possible processes in perturbation theory. As can be seen in the plot, this changes for higher orders. As for the ED case, we motivate the found scaling in Eq. (42) with the dominant second order perturbation theory between the two parts of the Hamiltonian.

All in all, the finite-size calculations are in perfect agreement with the findings of Sec. 2. While the results on finite systems differ from the effective model, the discrepancies shrink for larger $N$, allowing for an extrapolation in $N$ with a vanishing difference in the thermodynamic limit.

## 4 Conclusions

In this work we considered a class of Dicke models with a generalized matter part consisting of long-range hopping and correlated processes while keeping the simple form of the single light mode and the Dicke light-matter coupling. In the limit of weak light-matter interactions and large system sizes, we were able to establish an analytical solution of the low-energy physics by mapping the system to an effective conventional Dicke model without matter-matter interactions. To prove the presented mapping, we demonstrated that the commutator between the two parts of the Hamiltonian vanishes in the thermodynamic limit for the given prerequisites. With this mapping in place, we are able to determine low-lying excitations on general footing by performing a Bogoliubov transformation, including the closing of the gap indicating a potential second-order phase transition. This reasoning is exemplified for the paradigmatic Dicke-Ising chain, which is compared to mean-field [2] and quantum Monte Carlo [36] results as well as finite-size calculations using ED and pcst$^{++}$ [3], yielding a satisfying agreement with our found mapping.

Next, it would be interesting to generalize our findings further, both with a more complex matter or light part. This includes adding more light modes, be it in a discrete or continuous way [38, 47, 48], and matter-matter interactions that do not conserve the number of spin-flip excitations. The latter terms are naturally present in a variety of interacting quantum many-body systems relevant for quantum-optical and solid state platforms including paradigmatic examples like the transverse-field Ising or Heisenberg model. This generalization would give the mapping a wider range of potential applications to foresee the impact of light-matter interactions on well-discussed strongly correlated matter systems.

While these potential generalizations offer a broader range of systems to apply the mapping to, it would be also an interesting question to find the fundamental limits where this type of mapping is applicable. Introducing terms that prevent the mapping to hold would therefore allow to find emerging physical phenomena that can not be described by a non-interacting matter part in an effective model. These insights could be a building block toward tailored effective models by specific light-matter systems to induce new types of effective interactions.

# Acknowledgments

We thank Max Hörmann, Anja Langheld, and Jonas Leibig for fruitful discussions on the Dicke Ising model. We further thank Lea Lenke for helping to fine-tune some formulations of the work. We acknowledge the use of the following python libraries to compute and visualize the presented results: `NumPy` [49], `SciPy` [50], `SymPy` [51] and `Matplotlib` [52,53]. For the pcst$^{++}$ calculations we use the `pcstpp_CoefficientGenerator` [3,54].

**Author contributions**   AS: Conceptualization, Data curation, Formal analysis, Investigation, Methodology, Writing – original draft, Writing – review & editing. KPS: Conceptualization, Methodology, Supervision, Writing – review & editing.[1]

**Funding information**   This work was funded by the Deutsche Forschungsgemeinschaft (DFG, German Research Foundation) – Project-ID 429529648 – TRR 306 QuCoLiMa(Quantum Cooperativity of Light and Matter). KPS acknowledges the support by the Munich Quantum Valley, which is supported by the Bavarian state government with funds from the Hightech Agenda Bayern Plus.

**Data availability**   Supplementary data for all figures, including the analytical results and the obtained pcst$^{++}$ series expansions, are available online [44].

# A   Deriving the Hamiltonian in momentum space

In this section we show the derivation of Eq. (8) in greater detail and prove the bosonic commutation relation for the magnonic operators in momentum space in the thermodynamic limit. Starting with the Fourier transformation of Eq. (7) we can directly extract the inverse transformation as

$$b_j = \frac{1}{\sqrt{N}} \sum_k e^{ikj} \tilde{b}_k, \qquad b_j^\dagger = \frac{1}{\sqrt{N}} \sum_k e^{-ikj} \tilde{b}_j^\dagger. \tag{A.1}$$

Inserting Eq. (7) into above equation we get identity using the definition of the momentum on periodic lattices

$$\sum_k e^{ik(j_1 - j_2)} = N \delta_{j_1, j_2}. \tag{A.2}$$

We obtain Eq. (8) by inserting Eq. (A.1) into Eq. (1). In the following we will take a closer look at some of the terms for convenience. First, we can directly show $\sum_j b_j^\dagger b_j = \sum_k \tilde{b}_k^\dagger \tilde{b}_k$ by again using Eq. (A.2):

$$\sum_j b_j^\dagger b_j = \frac{1}{N} \sum_j \sum_{k,p} e^{ij(p-k)} \tilde{b}_k^\dagger \tilde{b}_p = \frac{1}{N} \sum_{k,p} \tilde{b}_k^\dagger \tilde{b}_p \underbrace{\sum_j e^{ij(p-k)}}_{=N\delta_{p,k}} = \sum_k \tilde{b}_k^\dagger \tilde{b}_k. \tag{A.3}$$

---

[1]Following the taxonomy CRediT to categorize the contributions of the authors.

Analogously, the transformation of the hopping terms in $\mathcal{H}_{\text{matter}}$ into momentum space can be shown. For the interaction processes in $\mathcal{H}_{\text{matter}}$, we again insert Eq. (A.1) obtaining

$$\sum_{j,\delta_1,\delta_2,\delta_3} c_{\delta_1,\delta_2,\delta_3} b_j^\dagger b_{j+\delta_1}^\dagger b_{j+\delta_2} b_{j+\delta_3}$$

$$= \frac{1}{N^2} \sum_{k_1,k_2,p_1,p_2} \sum_{j,\delta_1,\delta_2,\delta_3} c_{\delta_1,\delta_2,\delta_3} e^{ij(p_1+p_2-k_1-k_2)} e^{i\delta_2 p_1 + i\delta_3 p_2 - i\delta_1 k_2} \tilde{b}_{k_1}^\dagger \tilde{b}_{k_2}^\dagger \tilde{b}_{p_1} \tilde{b}_{p_2} \quad \text{(A.4)}$$

$$= \frac{1}{N} \sum_{k_1,k_2,p_1,p_2} \sum_{\delta_1,\delta_2,\delta_3} c_{\delta_1,\delta_2,\delta_3} e^{i\delta_2 p_1 + i\delta_3 p_2 - i\delta_1 k_2} \tilde{b}_{k_1}^\dagger \tilde{b}_{k_2}^\dagger \tilde{b}_{p_1} \tilde{b}_{p_2} \delta_{k_1+k_2,p_1+p_2} . \quad \text{(A.5)}$$

Next, we show the derivation of the bosonic particle statistics of the magnons in the thermodynamic limit. Therefore, we calculate the commutation relation of the magnonic operators in momentum space. We make use of the hardcore boson statistics of the spins 1/2 particles in real space

$$\left[ b_j, b_l^\dagger \right] = \delta_{j,l}(1 - 2n_j), \quad \text{(A.6)}$$

with $n_j = b_j^\dagger b_j$. Using Eq. (7) we obtain the commutator as

$$\left[ \tilde{b}_p, \tilde{b}_k^\dagger \right] = \frac{1}{N} \sum_{j,l} e^{-ipj+ikl} \left[ b_j, b_l^\dagger \right] \quad \text{(A.7)}$$

$$= \frac{1}{N} \sum_{j,l} e^{-ipj+ikl} \delta_{jl}(1 - 2n_j) \quad \text{(A.8)}$$

$$= \frac{1}{N} \sum_{j} e^{ij(k-p)}(1 - 2n_j) \quad \text{(A.9)}$$

$$= \delta_{p,k} - \frac{2}{N} \sum_{j} e^{ij(k-p)} n_j \quad \text{(A.10)}$$

$$= \delta_{p,k} - \frac{2}{N} \sum_{k'} \tilde{b}_{k'}^\dagger \tilde{b}_{k'+p-k} . \quad \text{(A.11)}$$

Restricting ourselves to finite particle numbers according to Eq. (6), the second term vanishes in the thermodynamic limit. We are left with the bosonic commutation relation $[\tilde{b}_p, \tilde{b}_k^\dagger] = \delta_{p,k}$. The remaining commutation relations for bosonic operators $[\tilde{b}_p, \tilde{b}_k] = [\tilde{b}_p^\dagger, \tilde{b}_k^\dagger] = 0$ follow directly from $[b_j, b_l] = [b_j^\dagger, b_l^\dagger] = 0$.

# B  Calculating the commutator norms

In this section we derive the result of $[\bar{\mathcal{H}}_{\text{D}}, \mathcal{H}_{\text{MM}}]$ vanishing in the thermodynamic limit in more detail. As mentioned in Subsec. 2.3 we do so by calculating an upper bound of the operator norm $\|\cdot\|$ for the given commutators. The operator norm $\|\mathcal{O}\|$ of an operator $\mathcal{O}$ is defined as

$$\|\mathcal{O}\| \equiv \sup_s \sqrt{\langle s|\mathcal{O}^\dagger \mathcal{O}|s\rangle}, \quad \text{(B.1)}$$

with $|s\rangle$ being any normalized state with $\langle s|s\rangle = 1$. For our discussion, we will restrict the choice of $|s\rangle$ onto those states that fulfill the prerequisites given in Eq. (6), namely $\langle s| \sum_j n_j |s\rangle < \infty$.

We start the derivation by showing that the norm of the hopping operator

$$\mathcal{O} = \sum_j b_j^\dagger b_{j+\delta}, \quad \text{(B.2)}$$

does not scale with the system size and thus stays finite in the thermodynamic limit for any chosen $\delta$. First, we use the definition of the norm in Eq. (B.1) to write it as an expectation value for the given operator

$$\|\mathcal{O}\|^2 = \sup_s \sum_{j,l} \langle s|b_l^\dagger b_{l+\delta}\, b_j^\dagger b_{j+\delta}|s\rangle \,. \tag{B.3}$$

While $s$ may be any valid state, let us for the moment consider an $n$-magnon state in real space of the form

$$|s\rangle = |j_1, j_2, \ldots, j_n\rangle \,, \tag{B.4}$$

where the $j_i$ denote a magnon at position $j_i$. Applying $\mathcal{O}$ on this state leaves us with a state of the form

$$\mathcal{O}|s\rangle = \sum_{s' \in \mathcal{S}_s} c_{s\to s'} |s'\rangle \,, \tag{B.5}$$

with $|c_{s\to s'}| \leq 1$ and $\mathcal{S}_s$ being a *finite* set of states $s'$ depending on the initial state $s$. We can find an upper bound of $|\mathcal{S}_s| \leq n$ because of the annihilation operator $b_{l+\delta}$ acting once on all $j_i$ in Eq. (B.3). As Eq. (B.5) holds for any state $s$ in real-space basis, we can conclude that

$$\mathcal{O}^\dagger\mathcal{O}|s\rangle = \sum_{s' \in \mathcal{S}_s} \sum_{s'' \in \mathcal{S}_{s'}} c_{s\to s'} c_{s'\to s''} |s''\rangle \,, \tag{B.6}$$

consists out of a maximum of $|\mathcal{S}_s| \cdot |\mathcal{S}_{s'}| \leq n^2$ terms independent of the system size with $|c_{s\to s'} c_{s'\to s''}| \leq 1$. We can therefore conclude that

$$\sqrt{\langle s|\mathcal{O}^\dagger\mathcal{O}|s\rangle} \leq n = \langle s|\sum_j n_j|s\rangle \,. \tag{B.7}$$

As $\langle \sum_j n_j \rangle$ has to stay finite according to our setup (see Eq. (6)), we can conclude that $\sqrt{\langle\mathcal{O}^\dagger\mathcal{O}\rangle}$ stays finite, too. Last, we generalize this statement to arbitrary states, being a (potentially infinite) superposition of the states of Eq. (B.4). Assuming a normalized eigenstate of $\mathcal{O}$ as

$$|v\rangle = \sum_s c_s |s\rangle \,, \quad \sum_s |c_s|^2 = 1 \,, \tag{B.8}$$

we conclude that the corresponding eigenenergy $e$ can be again bounded from above by $n$, as done for the restricted state in Eq. (B.7). Therefore, Eq. (B.7) also holds for arbitrary superposition of states. As a last step in preparation, we generalize the investigated hopping operator $\mathcal{O}$ from Eq. (B.2):

$$\mathcal{O} = \sum_j b_j^\dagger o(j) b_{j+\delta} \,, \quad o(j) := b_{j+\delta_1}^\dagger \cdots b_{j+\delta_{m_1}}^\dagger b_{j+\delta_{m_1+1}} \cdots b_{j+\delta_{m_1+m_2}} \,, \tag{B.9}$$

with $m_1 + m_2$ finite and arbitrary $\delta_{m_i}$ values. Even though the new operator offers a much more complicated structure, the above argumentation can be followed analogously, as the sums are again cut to a finite subset and $\|b_j^{(\dagger)}\| \leq 1$, leading to an upper bound of

$$\sqrt{\langle s|\mathcal{O}^\dagger\mathcal{O}|s\rangle} \leq \sqrt{n \cdot (n + m_1 - m_2)} \leq n + |m_1 - m_2| = \langle s|\sum_j n_j + |m_1 - m_2||s\rangle \,, \tag{B.10}$$

for the generalized operator. As $\langle s|\sum_j n_j|s\rangle \equiv n$ and $|m_1 - m_2|$ are bound to finite values, we can again conclude that the expectation value stays finite.

We now apply these findings to the commutators in Eqs. (14) and (16). To do so, we bring the expressions into the form of Eq. (B.9). We will show the derivation only for one

of the terms each, as the other can be derived analogously. We use Eq. (7), to rewrite the commutator Eq. (14) in in terms of real-space operators:

$$\sum_{k \neq 0} e^{ik\delta} [\tilde{b}_0, \tilde{b}_k^\dagger \tilde{b}_k] = \sum_{k \neq 0} \frac{1}{N^{3/2}} \sum_{j,l,m} e^{ik(l-m+\delta)} [b_j, b_l^\dagger b_m] \tag{B.11}$$

$$= \frac{1}{N^{3/2}} \sum_{l,m} (1 - 2n_l) b_m (N\delta_{l+\delta,m} - 1) \tag{B.12}$$

$$= -\frac{2}{\sqrt{N}} \left( \sum_l n_{l-\delta} b_l - \sum_l n_l \tilde{b}_0 \right). \tag{B.13}$$

For above calculation we have used $[b_j, b_l^\dagger] = \delta_{j,l}(1 - 2n_l)$ and $\sum_k e^{ik(l-m+\delta)} = N\delta_{l+\delta,m}$. When taking the norm of above commutator, we can use the identities $\|AB\| = \|A\|\|B\|$ and $\|A + B\| \leq \|A\| + \|B\|$ to obtain

$$\| \sum_{k \neq 0} e^{ik\delta} [\tilde{b}_0, \tilde{b}_k^\dagger \tilde{b}_k] \| \leq \frac{2}{\sqrt{N}} \left( \| \sum_l n_{l-\delta} b_l \| + \| \sum_l n_l \| \| \tilde{b}_0 \| \right). \tag{B.14}$$

As the operators $\sum_l n_{l-\delta} b_l$ and $\sum_l n_l$ fulfill the form of Eq. (B.9) their norms stay finite. For the remaining $\tilde{b}_0$ the same can be shown. So, the overall commutator scales with $N^{-1/2}$ and therefore vanishes in the thermodynamic limit.

We move on with the commutator in Eq. (16). We write down the commutator in real-space operators and use the respective commutation relations:

$$\sum_{j,l} [b_l, b_j^\dagger b_{j+\delta_1}^\dagger b_{j+\delta_2} b_{j+\delta_3}] = \frac{1}{\sqrt{N}} \sum_{j,l} \left( b_j^\dagger [b_l, b_{j+\delta_1}^\dagger] + [b_l, b_j^\dagger] b_{j+\delta_1}^\dagger \right) b_{j+\delta_2} b_{j+\delta_3} \tag{B.15}$$

$$= \frac{1}{\sqrt{N}} \sum_j \left( b_j^\dagger (1 - 2n_{j+\delta_1}) + b_{j+\delta_1}^\dagger (1 - 2n_j) \right) b_{j+\delta_2} b_{j+\delta_3}. \tag{B.16}$$

Eq. (B.16) can be expressed in terms of four operators of the form of Eq. (B.9), leading to the conclusion that the overall commutator scales with $N^{-1/2}$.

To sum up this section, we have introduced the operator norm as a suitable measure to show the decoupling of $\bar{\mathcal{H}}_{\text{D}}$ and $\mathcal{H}_{\text{MM}}$ in the thermodynamic limit by finding an upper bound for a general operator $\mathcal{O}$ in Eq. (B.9) and tracing back the commutator $[\bar{\mathcal{H}}_{\text{D}}, \mathcal{H}_{\text{MM}}]$ to terms of this form.

## C  Solving the Dicke model

In this section we derive the solution of the effective Dicke models occurring in the main text, namely Eqs. (9), (25), and (37). To keep this derivation concise, we will not cover all technical details needed for the transformations to work. For a more technical and detailed work on the Bogoliubov transformation itself, we recommend the paper by Xiao [55]. For this section we will stick closely to the formulation in this paper.

The general starting point for all three Hamiltonians referenced above is their quadratic nature, namely that all terms consist out of exactly two operators apart from a constant term. All three Hamiltonians can therefore be rewritten in a general form as

$$\mathcal{H} = \sum_{i,j} \alpha_{i,j} b_i^\dagger b_j + \frac{1}{2} \gamma_{i,j} b_i^\dagger b_j^\dagger + \frac{1}{2} \gamma_{j,i}^* b_i b_j, \tag{C.1}$$

where $b_i^{(\dagger)}$ are annihilation (creation) operators with pairwise bosonic commutation relations and $\alpha_{i,j}, \gamma_{i,j} \in \mathbb{C}$. Note that these commutation relations only hold in the thermodynamic limit for our models (see App. A) as well as for the bare Dicke model [42]. Thus, the presented solution only applies in the limit $N \to \infty$, which is nonetheless our point of interest.

For this class of models Bogoliubov introduced a linear transformation to diagonalize these systems [56]. While the transformation was also generalized to particles with fermionic commutation relations [57–59], we can limit ourselves to the bosonic case here. It is convenient to write the Hamiltonian of Eq. (C.1) in matrix notation as

$$\mathcal{H} = \frac{1}{2}\psi^T M \psi - \frac{1}{2}\mathrm{tr}(\alpha), \qquad M = \begin{pmatrix} \alpha & \gamma \\ \gamma^\dagger & \alpha \end{pmatrix}, \qquad \psi = \begin{pmatrix} b \\ b^\dagger \end{pmatrix}, \qquad \text{(C.2)}$$

where $\alpha$, $\gamma$ are matrices built out of the scalars $\alpha_{i,j}$, $\gamma_{i,j}$ and $b$, $b^\dagger$ are vectors with the annihilation and creation operators $b_i$, $b_i^\dagger$, respectively, as their entries.

The Hamiltonian is solved by introducing a linear transformation of the form

$$b = Ad + Bd^\dagger, \qquad \text{(C.3)}$$

with $d^{(\dagger)}$ being vectors, analogous to $b^{(\dagger)}$, consisting out of new bosonic operators $d_i$, $d_i^\dagger$ and coefficient matrices $A$, $B$. These matrices have to be chosen such that they diagonalize $\mathcal{H}$ and at the same time preserve the desired bosonic commutation relations for the $d_i^{(\dagger)}$ operators, which is not guaranteed to be possible. In [55] an equivalence between the existence of such a transformation and the diagonalization of a so-called dynamical matrix

$$D \equiv \begin{pmatrix} I & 0 \\ 0 & -I \end{pmatrix} \cdot M, \quad \text{with } I \text{ being the identity matrix,} \qquad \text{(C.4)}$$

is proven, which is advantageous for actual calculations. If and only if $D$ is diagonalizable with all its eigenvalues being real, the Bogoliubov transformation can be applied. To keep this section concise, we refer to Xiao for the derivation of this finding [55]. If the transformation exists, the resulting Hamiltonian can be written as

$$\mathcal{H} = \sum_i \omega_i d_i^\dagger d_i + C, \qquad \text{(C.5)}$$

with $\omega_i$ being the eigenenergy of the $i$-th elementary excitation and a constant term $C$. The eigenvalues $\omega_i$ are the positive eigenvalues of $D$ if the transformation exists.

For our models discussed in the main text, we find that the transformation is applicable for the case of small light-matter couplings. The transformation starts failing when the respective gap of the normal phase closes. The Hamiltonian in Eq. (25) can be written in matrix notation in terms of Eq. (C.2) as

$$M = \begin{bmatrix} \omega & g & 0 & g \\ g & \omega_0 - 2cJ & g & 0 \\ 0 & g & \omega & g \\ g & 0 & g & \omega_0 - 2cJ \end{bmatrix}, \qquad \psi = \begin{pmatrix} a \\ \tilde{b}_0 \\ a^\dagger \\ \tilde{b}_0^\dagger \end{pmatrix}. \qquad \text{(C.6)}$$

Diagonalizing the dynamical matrix yields the positive eigenvalues of Eq. (27). The general Hamiltonian of Eq. (9) can be solved analogously. For the Hamiltonian of the antiferromagnetic phase in Eq. (37), we have to take into account three types of quasiparticles, resulting in

the $6 \times 6$ matrix

$$M = \frac{1}{\sqrt{2}} \begin{bmatrix} \sqrt{2}\omega & g & g & 0 & g & g \\ g & a & 0 & g & 0 & 0 \\ g & 0 & b & g & 0 & 0 \\ 0 & g & g & \sqrt{2}\omega & g & g \\ g & 0 & 0 & g & a & 0 \\ g & 0 & 0 & g & 0 & b \end{bmatrix}, \qquad \psi = \begin{pmatrix} a \\ \tilde{b}_{0,A} \\ \tilde{b}_{0,B} \\ a^{\dagger} \\ \tilde{b}_{0,A}^{\dagger} \\ \tilde{b}_{0,B}^{\dagger} \end{pmatrix}, \qquad \text{(C.7)}$$

with $a/\sqrt{2} = 2cJ - \omega_0$, $b/\sqrt{2} = 2cJ + \omega_0$, which can be solved with a computer algebra program. As the eigenvalues can not be written down in a concise form, we omit to write them down here [44].

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
