# Peer review of "(Almost) Everything is a Dicke model -- Mapping non-superradiant correlated light-matter systems to the exactly solvable Dicke model"

_SciPost Physics Core, doi:SciPost Phys. Core 7, 038 (2024)_

## Round 3 · Referee Report · Anonymous · 2024-5-6

Report

The revised manuscript seems now suitable for publication in SciPost Physics Core. The clarity of the manuscript has been improved, and the scope of the results is now made sufficiently clear. The work is useful for those in related fields, and as noted in my previous report, it helps explain some features that have been observed in numerical results.

I note below one very minor point regarding formatting that the authors may wish to fix.

Requested changes

Equations 23 and 33 should probably have parentheses added in the light matter coupling term to match the form of Eq. 3.

Recommendation

Publish (meets expectations and criteria for this Journal)

---

## Round 3 · Referee Report · Anonymous · 2024-5-15

Report

The remarks of both referees have been fully addressed and the resulting article is now fully meeting the Journal's acceptence criteria.

I, consequently, recommend it be published in SciPost Physics Core.

Recommendation

Publish (meets expectations and criteria for this Journal)

---

## Round 3 · Author Response

Errors in user-supplied markup (flagged; corrections coming soon)

# Reply to first referee

*We thank the Referee for their thorough examination of our manuscript and the positive evaluation. Below we address the individual proposed changes raised by the referee:*
* * *
1. In the introduction, the selection of papers about various extensions of the Dicke model does not seem well structured. For example, some of these papers are on situations with multiple photon modes (e.g. Refs 16, 25) but no distinction is made between single and multimode experiments. The field of extensions of the Dicke model is very large so it is not reasonable to cite everything in this field. However the references given should match better to the particular extensions being listed.

*Our answer: We have added new categories (namely multi-mode cavities and non-equilibrium systems) to provide a better distinction in the overview of literature. We have further concentrated in the citations on prime examples of the respective category, to hopefully provide a better starting point for the reader.*
* * *
2. Equation 3 should probably have parentheses in the last term.

*Our answer: We changed the equation as proposed.*
* * *
3. The statements in Eq. 5 and 6 could be made more precise. I think the meaning assumed here is that all these quantities approach a finite limit as N goes to infinity; if so, this should be stated more clearly.

*Our answer: This is completely correct. We have added a sentence around the two equations explicitly stating that we operate in the thermodynamic limit.*
* * *
4. After Eq. 10, when saying the two parts decouple in the thermodynamic limit, I think that this should also say it assumes finite occupations (i.e. assumes Eq. 6). This restriction is noted in all other cases where the thermodynamic limit is discussed.

*Our answer: We have added your proposed statement. Furthermore, following the proposal of the other report, we explicitly stated that the statement holds only in the non-superradiant phases.*
* * *
5. In Figure 2 caption, the blue region (hybrid magnon-photon continuum) is not explained that clearly. The wording here describes this as the "two-magnon subspace ... which is modified by the hybrid state at k=0". However since the key point of the paper is that the Dicke sector and Magnon sector decouple, this wording seems surprising. I assume what is meant here is that this continuum is formed of states that contain one excitation in the Dicke sector and one in the magnon sector. This needs clarifying.

*Our answer: Your assumption is totally correct. As proposed, we have added an explanation of this blue region in the caption of Figure 2.*
* * *
6. On page 6, start of Sec. 2.3, "we proof" should read "we prove"

*Our answer: We corrected the error.*
* * *
========================================

# Reply to second referee

*We thank the referee for their thorough examination of our manuscript and the well argued possibilities to improve. Below we address the individual proposed changes raised by the referee:*
* * *
1- One the issues which should be addressed is that the abstract, title and overall writing of the early sections of the article definitely seem to present the results as much broader reaching than they actually are.

A central limitation of the work is that it cannot capture the superradiant phase(s) which is certainly one of the most central aspect of Dicke-like model. The abstract and early section tend to talk exclusively about low energy sectors which naturally lead the reader to believe that the mapping should work over the full phase diagram by capturing the ground state and low energy excitation spectrum.

It would be necessary to mention early on in the paper and in the abstract that they are not really working in a low energy sector but in a low excitation number sector. The authors should be explicit about the fact that their work will exclude the treatment of any superradiant phases.

*Our answer: Thanks for your comment. We understand your concern that it can potentially be misleading for the reader. We have addressed your point in title, abstract, and the first sections to clarify the limitations of this mapping in a better way.*
* * *
- I also strongly feel that in view of this limitation, the title is also misleading and should therefore be changed to avoid making the vastly overreaching statement that it currently makes. The first part of that title “(Almost) everything is a Dicke model …” certainly does not convey the correct message about the content of the article and the strong limitation that excludes superradiant phases.

*Our answer: We have explicitly added the 'non-superradiant' term to the title. We have done this in the second part of the title, as this part explains in short the setup and limitations. So, we hope it gets clear from the beginning that we are not able to capture the superradiant phases.*
* * *
- The abstract does state:

Coming from the limit of weak light-matter couplings and large number of matter entities, we map the relevant low-energy sector of a broad class of models onto the exactly solvable Dicke model.

but that is insufficient to clearly lead the reader to understand the limitation. The abstract should make it clear that the phrase “Coming from the limit of weak light-matter couplings“ is to be understood as excluding superradiance.

*Our answer: We have added again explicitly the 'non-superradiant' term to make clear that the mapping only works in these phases. The same we have added analogously in the introduction.*
* * *
On page three the exact restriction imposed is finally introduced for the first time in eq. (6):

“For our findings we further restrict ourselves onto the low-energy subspace demanding :”

This, again doesn’t not feel completely accurate, since in general there can be an important difference between low-energy and low-excitation number. It think it should be made clear here again at this point in the text that superradiant phases cannot be treated.

*Our answer: We have added a statement at the beginning of this section that we only consider the non-superradiant phases. With this restriction the low-energy subspace does only refer to the subspace within the non-superradiant phases.*
* * *
I know that the sentence: “We focus on the non-superradiant phase with ω0,ω ≫ g inducing two quasiparticle types, namely photons and magnons” appears at the beginning of section 2.1 but the word “focus” is not strong enough to understand that the work explicitly excludes the SR phases

*Our answer: We have made this clearer by replacing 'focus' with 'restrict'.*
* * *
Only at the beginning of section 3 do we explicitely read: “the two non-superradiant phases of the model, which obey the prerequisites in Eq. (6) “ which finally makes the exclusion clearer.

Overall, the authors should review the first few sections, the abstract and the title to make the non-superradiant point clear, from the very start.
* * *
2- Another limitation of the mapping seems to be in the anti-ferromagnetic case. Here the authors simply state that they choose a bipartite lattice “To avoid any kind of geometric frustration”. They, however, should explicitly discuss whether or not the restriction to bipartite lattices is a requirement of the mapping or not.

From the way the mapping is then implemented, it seems to be an absolute necessity to have a bipartite lattice. If it is indeed the case, the authors should be explicit about it. If not, the authors need to comment on it because the way the result is presented clearly seems to imply it is necessary.

*Our answer: You are totally right. To make the reasoning clearer we have added a short discussion why the bipartite lattice is crucial for the antiferromagnetic case.*
* * *
3- There are multiple imprecise statements throughout the article concerning first and second-order phases transitions. Considering that the authors also keep making references to their not yet published article [43], it is essential that any statement relating this work to which transitions are first and second order, or to the results of [43], be more precise.

*Our answer: We have given more information about the limiting cases to get a qualitative picture of the locations of the first and second order regimes. For this we also refer to the paper [1] that investigated the limiting case of $\omega_0=0$. This finding also motivates that around this limit one can expect first-order phase transitions. Reference [43] is therefore a confirmation of this line of argumentation using QMC and at the same time shows (as [1] already did) that the mean-field results in [2] and our new results can not fetch the first order phase transitions, as discussed in the main text.*
* * *
Here are some example sentences:

. According to this mean-field treatment, all quantum phase transitions between the four phases are of second order. In contrast, recent works using perturbation theory [1] and quantum Monte Carlo [43] have found that "some of the transitions" are indeed of first order in 1D.

*Our answer: We have added a short discussion about the limiting case $\omega_0=0$ that was discussed in [1] and the argumentation why the first order phase transition survives around this limiting case.*
* * *
. Therefore, the first-order nature of "certain phase transition lines" observed by exact diagonalization [1] and quantum Monte Carlo calculations [43] is in no contradiction to our findings.

*Our answer: We added the qualitative statement that the transitions around the limit $\omega_0=0$ are of first order.*
* * *
. "In the realm of a second-order phase transition" the results of [2,43] and the presented effective theory do again coincide.

. In comparison to [43], the results agree for the "region of the second-order phase transition", while we are not capable of detecting the first-order phase transition found with quantum Monte Carlo calculations, as expected.

*Our answer: We added the limit $\omega_0=0$ to clarify the region of first and second order phase transitions.*
* * *
. Additionally, the ED results can be affected by the nearby phase transition, potentially happening before the closing of the gap by a first-order transition.

Most of these sentences are confusing to me as a reader. I think it should be made explicit which transitions are first and second order in the full phase diagram.

In the previous sentences, I added quotes around the many vague statements which fail to inform clearly the reader of which transition are first and second order. Words like “some of the transitions” or “certain phase transition lines” make it very hard especially without access to [43] to understand which are are which. The authors should be much more precise in their description of the various transition saying explicitly which one has been shown to be of which order.

The phrases “in the realm of a second-order transition” is used or “the results agree for the region of the second-order phase transition”, do not allow the reader to understand which specific region of the phase diagram the authors are talking about.

Therefore, they should be explicit about the order of every phase transition in figure 3 and an important effort of rewording of the various mentions of phase transitions has to be made in order to clarify the discussions and how it all relates to the unpublished results of [43].

*Our answer: We have named the realms of first- and second-order phase transitions qualitatively. We have not altered the phase diagram of Fig. 3, adding quantitative results from [43]. This will be done in a detailed way in [43].*
* * *
4- The whole physical intuition section 2.2 seems discusses some points ahead of the proof and in itself is not necessarily a bad idea. However, the sentence:

As only the k = 0 mode couples to the light field, it is reasonable that Eq. (11) will decouple from the light part in HD for N → ∞.

seems to be the central point which lead the reader to gather physical intuition about their main result (namely this decoupling). I, however, do not see why the fact fact that only the k=0 mode is coupled should somehow resonably lead to such a decoupling. The sentence seems to suggest that this is obvious, but if so, it really needs to be better explained why one should expect eq. 11 (which does contains k=0 terms) to decouple. If there is no such clear intuitive way of seeing it, the authors should replace “it is reasonable that “ by a formulation such as “it will be proven that” …

*Our answer: Thanks for your remark! This statement we did not make clear enough. We have now added a description how to understand this effect. We argue that the low-energy eigenstates of $H_{MM}$ are localized in real space and are thus broadened in momentum space. As the light-matter coupling only happens at a single frequency, we argue that the overlap of the bound state with $k=0$ vanishes in the thermodynamic limit. But of course this can only be a motivation.*
* * *
5- The authors say:

"As HMM conserves the number of magnons, this part of the model is block diagonal with respect to the number of particles and can be solved, e.g., with exact diagonalization as done in [46]. "

however, ref. 46 deals with the linked coupled cluster series expansion and not with what I would usually call exact diagonalisation methods. It might be worth clarifying this issue.

*Our answer: Thanks for pointing this out. We have deleted the half sentence and the reference to avoid any kind of confusion, as it should be understood only as a technical detail that is not crucial to the paper. The idea behind [46] is that the 'effective Hamiltonian' preserves the total particle number (as $H_{MM}$) and can than be diagonalized in the higher particle channels in a mixed momentum representation. We consider this as one possible way to diagonalize $H_{MM}$ in the individual particle channels.*
* * *
On the same topic, When the authors present their results in the last sections (figures 4 an 5) they never explicitly indicate how the energy of the g independent magnon modes, coming from the matter-matter parts of the hamiltonian, are computed. The answer might be very simple but I don’t see clearly the answer when looking at (28) for example even without the (ω0 −2cJ) ̃b0† ̃b0 term.

*Our answer: We have added the explicit formulars of the g-independent magnon modes with a bit of description.*
* * *
6- Some of the formulations are problematic either for a grammatical/syntactic point of view, some of which are liste d below and should be addressed:

a- Page 2: “ While the matter part of the Dicke model consists out of an arbitrary number of spins, it breaks down to a local problem in the case of no light-matter interaction, as the local degrees of freedom are only coupled through the cavity, making it trivially solvable. “

Since local is usually employed even for problems with short range interaction, I feel like the authors should modify the sentence to use “non-interacting problem” instead of “local problem”.

b- page 3: To keep the sums over all distances finite, we restrict the coefficients to hold

the verb to hold is not correct in this context and could be replaced by “to be such that” for exemple

c - page 5: Despite the correlated processes proportional to cδ1,δ2,δ3, the general Hamiltonian H in Eq. (1) is solely made up of uncorrelated one-magnon terms in the matter part, including number operators and hopping processes.

In this sentence the word despite is wrongly used. It should be “apart from” or “to the exception of” for exemple; using despite means that the correlated processes are also uncorrelated.

d- page 14-15: “While we have no direct handle to the deviation of the energies of the finite systems in relation to the derived effective theory, we can motivate the scaling of N−1 in Eq. (40) with perturbaion theory. “

[…]

“This motivates the found scaling of the energy in Eq. (40) “

the phrase motivate the scaling is used twice in three sentences making the second time sound bizarre (or at least redundant)

e- page 6: we proof in this concluding derivation section

proof -> prove

f- page 8: as it behaves trivial in this limit.

trivial -> trivially

g- page 9 unaffected of perturbations in g

unaffected of -> unaffected by

h- page 15: While this issue can be overcome by modifying the underlying transformation

overcome -> overcomed

i- Page 16: The sentence: Introducing terms that prevent the mapping to hold would therefore act as a potential candidate to find emerging physical phenomena that can not be described by a non-interacting matter part in an effective model.

reads awkwardly: Introducing terms […] would allow to find emerging … (would, for exemple, be a better wording than: introducing terms […] would act as …)

j -In appendix c:

“of exactly two operators despite a constant term.”
here again it appears that the word despite is wrongly used in place of “apart from” or “except for”

*Our answer: We have addressed all mentioned issues.*

---

## Round 3 · List of Changes

## Title
- Add 'non-superradiant' to title

## Abstract
- Add 'in the non-superradiant phases' to clarify the scope of the mapping.

## Introduction
- 2nd paragraph Change 'local' to 'non-interacting'
- 3rd paragraph Add new categories for generalized Dicke models, reduced the number of citations to only have the best fitting
- 4rd paragraph Change 'Focusing on the phase that is' to 'Restricting ourselves to the non-superradiant phases that are'
- 4rd paragraph Add 'in the non-superradiant phases'

## Derivation of the effective Dicke model
- 2nd paragraph Change 'focus on' to the stronger 'restrict ourselves to'
- Eq.(3) Add brackets around (b_j^\dagger + b_j)
- Above Eq.(5) Change 'hold' to 'stay finite in the limit of $N\to\infty$ as'
- Above Eq.(6) Add half sentence about the meaning of Eq. (6) to only have finite expectation values in the thermodynamic limit
- Last p. of 2.1 Add concluding remark that we have to stay in the non-superradiant phase
- 2nd p. of 2.2 Change 'Despite' to 'Apart from'
- 2nd p. of 2.2 Change 'despite' to 'apart from'
- P. below Eq.(11) Add short line of argumentation why the bound states decouple from the light field
- Fig.2 Add description of the blue hybridized continuum
- Last p. of 2.2 Remove reference to the Knetter paper
- First p. of 2.3 Change 'proof' to 'prove'

## Application onto the Dicke-Ising model
- First p. of 3.1 Discuss the qualitative realms of first and second order phase transitions as indicated by [1] and strengthen by the Monte Carlo study
- Below Eq.(18) Change 'trivial' to 'trivially'
- Below Eq.(21) Change 'of' to 'by'
- P. below Eq. (28) Describe and provide the energy of the (anti)-bound state
- P. below Eq. (29) Change text to have a clearer formulation where the first and second order phase transition takes place
- First p. of 3.3 Add sentence why the bipartite lattice is crucial for the antiferromagnetic case
- P. around Eq.(39) Describe and provide the energy of the (anti)-bound state
- Last p. of 3.3 Change text to have a clearer formulation where the first and second order phase transition takes place
- Last p. of ED Change 'motivates' to 'explains'

## Conclusions
- Last paragraph Change 'act as a potential candidate' to 'allow'

## Appendix
- 2nd paragraph Change 'despite' to 'apart from'

---

## Editorial Decision

published